# OpenReview forum: "URDFormer: Constructing interactive Realistic Scenes from Real Images via Simulation and Generative Modeling"
_robot-learning.org/CoRL/2023/Workshop/TGR — CoRL 2023 Workshop TGR Oral_

### Official Review · Reviewer_ctys · 2023-10-19

**Rating:** 9
**Confidence:** 5

**Review:**

The proposed method presents a valuable pipeline for scaling up simulation scenes, assets and training data, and closely aligns with the direction of diverse skill learning.

---

### Official Review · Reviewer_ppP4 · 2023-10-19
**Strong accept**

**Rating:** 8
**Confidence:** 4

**Review:**

Very nice result over real-to-sim URDF generation for articulated objects. It could be even better to include more discussion over how such system helps with robot control.

---

### Decision · Program_Chairs · 2023-10-20

**Decision:**

Accept (Oral)

**Comment:**

Great paper and closely aligned topic!